# Synthesis, Structural Characterization, and Biological Activity of New Pyrazolo[4,3-*e*][1,2,4]triazine Acyclonucleosides

**DOI:** 10.3390/molecules25010221

**Published:** 2020-01-05

**Authors:** Mariusz Mojzych, Zofia Bernat, Zbigniew Karczmarzyk, Joanna Matysiak, Andrzej Fruziński

**Affiliations:** 1Institute of Chemistry, Siedlce University of Natural Sciences and Humanities, ul. 3 Maja 54, 08-110 Siedlce, Poland; zosiabernat@wp.pl (Z.B.); zbigniew.karczmarzyk@uph.edu.pl (Z.K.); 2Department of Chemistry, University of Life Sciences, Akademicka 15, 20-950 Lublin, Poland; joanna.matysiak@up.lublin.pl; 3Department of General and Ecological Chemistry, Technical University, ul. Żeromskiego 115, 90-924 Łódź, Poland; andrzej.fruzinski@p.lodz.pl

**Keywords:** acyclonucleosides, pyrazolo[4,3-*e*][1,2,4]triazine, anticancer activity, X-ray analysis, theoretical calculation, molecular docking

## Abstract

A series of new pyrazolo[4,3-*e*][1,2,4]triazine acyclonucleosides **2**–**5** and **8** were prepared and evaluated for their anticancer activity against human cancer cell lines (MCF-7, K-562) and CDK2/E, as well as Abl protein kinases inhibitors. Lipophilicity of the compounds was determined using C-18 and immobilized artificial membrane (IAM) chromatography. In order to confirm the molecular structures and synthesis pathway of new acyclonucleosides, X-ray analysis was performed for model compound **3**. Theoretical calculations at the DFT/B3LYP/6-311++G(d,p) level were used for the characterization of electronic structures of **1**–**8**. The potential antiviral activity of acyclonucleosides **2**–**8** was tested in silico using molecular docking method.

## 1. Introduction

The discovery and use of aciclovir (**A**) [1] in medicine as a useful and selective antiviral therapeutic agent, and the development of viral diseases has made the synthesis of new acyclic nucleosides a relevant and very interesting subject of study. The modification of the guanine moiety and the acyclic side chain in aciclovir generated several new compounds of significant antiviral activity, e.g., ganciclovir (**B**) [2], penciclovir (**C**) [3], and famciclovir (**D**) [4] (Figure 1). To increase the oral bioavailability of acyclonucleosides, some of them were converted to prodrug forms using valine for esterification. Known in the medical drugs of this group are: valganciclovir (**E**) [5] and valaciclovir (**F**) [6] (Figure 1).

Despite the many research papers devoted to the new acyclonucleosides, very little information is given in the literature about acyclonucleosides with a pyrazolo[4,3-*e*][1,2,4]triazine ring system [7,8], which can be considered as bioisosteric to purine with a wide range of biological activities such as antimicrobial, antifungal, antitumour, antiproliferative, antiviral, and activity against HSV-1.

Naturally occurring pyrazolo[4,3-*e*][1,2,4]triazine derivatives are produced by the microorganisms and show anticancer and antibacterial activity [9,10,11]. Based on our earlier results related to synthesis and functionalization of pyrazolo[4,3-*e*][1,2,4]triazine core [12,13] and continuing our work on the preparation of acyclonucleosides [7,8]. Herein, we will discuss the synthesis, structure, and biological activity of new acyclonucleosides containing a pyrazolo[4,3-*e*][1,2,4]trazine moiety. A key feature of this strategy is outlined in Scheme 1.

## 2. Results and Discussion

### 2.1. Chemistry

The starting 3-methyl-5-methylsulfanyl-1*H*-pyrazolo[4,3-*e*][1,2,4]triazine (**1**) was synthesized from 5-acetyl-3-methylsulfanyl-1,2,4-triazine in a one-pot reaction by condensation with hydrazine hydrochloride, followed by the acid-promoted ring closure of the resulting intermediate [14]. The target acyclonucleosides **2**–**5** were prepared using previously described methods for the synthesis of *N*-alkylated 1*H*- and 2*H*-pyrazolo[4,3-*e*][1,2,4]triazine derivatives [14]. The synthesis was achieved by the reaction of **1** with appropriate alkylating agents in the presence of potassium carbonate in DMF stirring the reaction mixture, at room temperature overnight. The isomeric products were isolated by column chromatography. Compound **8** was synthesized in three steps from 3-methyl-5-methylsulfanyl-1*H*-pyrazolo[4,3-*e*][1,2,4]triazine (**1**). First, derivative **1** was reacted with dibromoethane in the presence of potassium carbonate in DMF to give as a main product **6** with 46% yield and small amount byproducts **6a** and **6b.** In the next step, pyrazolotriazine **6** was reacted with diethylmalonate and undergo nucleophilic substitution reaction to yield intermediate **7** in good yield. The treatment of the resulting 1*H*-pyrazolo[4,3-*e*][1,2,4]triazine **7** with sodium borohydrate in mixture of MeOH and CH_2_Cl_2_ in the ratio 1:1 gave the final analogue of penciclovir **8** and derivative **8a** as a minor product in 7% yield. The newly synthesized compounds were characterized by ^1^H NMR spectroscopy, HR-MS mass spectrometry, and elemental analysis. The spectral data confirmed that all of the new compounds had the expected structures and were of high purity.

### 2.2. Biological Activity

The cyclin-dependent kinases (CDKs) are a family of Ser/Thr kinases which, in association with specific cyclins, play critical roles as regulators of the different phases of the cell cycle. These enzymes and their direct regulators are frequently mutated, amplified, or deleted in malignant cells, suggesting that pharmacological CDK inhibition may be an effective strategy for treating cancer [15].

The Bcr-Abl chimeric protein is thought to play a central role in the pathogenesis of Philadelphia (Ph) chromosome-positive leukaemia, notably chronic myeloid leukaemia (CML) [16]. This abnormality is due to the reciprocal translocation between chromosomes 9 and 22. Three fusion proteins can be formed as a result of breakpoint in Bcr, all of which exhibit deregulated PTK activity [17,18,19]. Basic mechanisms that have been attributed to Bcr-Abl positive cells, particularly in CML, are increased proliferation, increased resistance to apoptosis [20,21,22], and an alteration of their adhesion properties [23,24]. Mutational analysis show that the tyrosine kinase activity of the protein is an absolute requirement for malignant transformation, and that it cannot be complemented by any downstream effectors [25,26]. For these reasons, an inhibitor of the Bcr-Abl tyrosine kinase should be an effective and selective treatment for CML. Selective therapies are aimed for the treatment of CML because its target is well defined in contrast to other cancers of body [27]. Hundreds of protein kinases are known in the human genome and a drug was required that targeted a single ATP binding site of protein kinase [28]. By blocking the binding of ATP, phosphorylation is prevented and Bcr-Abl expressing cells either have a growth disadvantage or they undergo apoptosis [22].

With respect to our recent studies on the anticancer activity of sulfonamide derivatives of pyrazolo[4,3-*e*][1,2,4]triazines observed in vitro in cell lines expressing fusion oncoprotein Bcr-Abl [29], we have evaluated the cytotoxicity of the new acyclonucleosides against two human cancer cell lines, namely breast adenocarcinoma-derived MCF7 and chronic myelogenous leukemia K562. The results presented in Table 1 demonstrate that none of the compounds possessed cytotoxicity within the tested concentration range. In addition, we have also assayed the new compounds for the inhibition of protein kinases CDK2/cyclin E and Abl as described previously [29]. However, in contrast to previously described 5-anilino-pyrazolo[4,3-*e*][1,2,4]triazines that inhibited Abl kinase in micromolar concentrations, none of the newly tested compounds proved the ability to inhibit any kinase (Table 1). This could be due to the lack of a NH group between the heterocyclic core and the phenyl ring that is responsible for binding to the kinase.

### 2.3. Lipophilicity

Reversed-phase high performance liquid chromatography (RP HPLC) was used to determine the lipophilicity of compounds. The two the biomimetic stationary phases, octadecyl (C-18) and phosphatidylcholine (immobilizer artificial membrane, IAM) were applied. The UV-Vis spectra of all compounds were measured in acetate buffer–methanol solutions of different pH values (pH = 4 and 7.4). Spectra of compounds: (**1**) parent, unsubstituted compound and (**2**) are presented in Figure 2. These spectra show that the pH of solution did not influence the electron structure of the studied compounds (Appendix A). The analysis of the microspecies distribution of the compounds depending on the pH (in the range of 4–7.4) using numerical methods also indicates the lack of pH influence on the structure of the compounds (MarvinViw 5.9.4). As pH = 7.4 is recommended for IAM chromatography measurements, other chromatographic studies were performed at this pH (physiological pH). For comparison, C-18 chromatography was performed at pH = 4.

The regular retention changes of compounds in the function of the organic modifier in the mobile phase was found for both RP-18 and IAM chromatography. Such dependence is expressed by the Soczewiński-Wachtmeister equation [31]:log k = log k_w_ + S (% organic modifier)(1)
where log k_w_—the intercept and *S*—slope of the regression curve.

Log k_w_ expresses the retention factor of a solute with pure water as the mobile phase. The parameters: log k_w_ obtained by extrapolation technique and S are commonly used as lipophilicity descriptors [32,33,34,35]. The obtained data are presented in Table 2 and Table 3. The dependence between the log k_w_ values obtained using the octadecyl stationary phase for different pH mobile phases is expressed by Equation (2).
log k_w(7.4)_ = 0.698 (± 0.384) log k_w(4)_ + 0.067 (± 0.699)(2)
*n* = 7; R = 0.902; s = 0.278; F = 21.9; *p* = 0.005; Q^2^ = 0.501

Compound **1** is the outlier. The relationship between the log k_w_ values obtained using the IAM and C-18 stationary phases at pH = 7.4 is described by the equation:log k_w(7.4)_ = 1.434 (± 0.392) log k_w(IAM)_ + 0.310 (± 0.313)(3)
*n* = 8; R = 0.964; s = 0.170; F = 380.03; *p* = 0.0001; Q^2^ = 0.885

Some molecular descriptors of compounds were estimated in silico (Table 3). The molar refractivity (MR) and log P values were calculated according to the fragmentation method introduced by Crippen [36]. For comparison, Clog P was determined. The data show that some compounds possess a low distribution coefficient expressed by log P as well as by log k_w_ values determined on phosphatidylcholine as the stationary phase. It can be assumed that the low lipophilicity of compounds (Clog P < −0.1) will not be conducive to their biological potency. This may limit absorption and distribution processes.

### 2.4. X-ray Analysis

The structure of the synthesized acyclonucleosides was unambiguously established by X-ray crystallography, taken 2-(2-hydroxyethyl)-3,5-dimethyl-2*H*-pyrazolo[4,3-*e*][1,2,4]-triazine (**3**), as a model compound. Crystal structure determination of (**3**) confirmed its assumed molecular structure and synthesis pathway. The structure of compound (**3**) is shown in Figure 3. The bond distances and angles in molecule of (**3**) are in normal ranges and they are comparable to the corresponding values observed in closely related structure of 1,3-dimethyl-5-methylsulfonyl-1*H*-pyrazolo[4,3-*e*][1,2,4]triazine [37]. The 2-hydroxyethyl substituent has a gauche-gauche conformation with the torsion angles N7-N8-C10-C11 of 82.0(2)° and N8-C10-C11-O1 of 61.3(2)°. The methylthio substituent lies almost in the plane of the triazine ring with the torsion angle N2-C3-S1-C13 of 179.98(15)°.

The packing of the molecules in the crystal structure of (**3**) is governed by O1-H1…N2*^i^* intermolecular hydrogen bond, linking the molecules into molecular chains parallel to the direction; O1-H1 = 0.86(3), H1…N2 = 2.07(3), O1...N2 = 2.928(2) Å, O1-H1…N2 = 176(3)^o^ and (*i*) = ¼ + *x*, ¼ − b*y*, −3/4 + *z*. Moreover, the *π*-electron systems of the pairs of pyrazole and triazine rings belonging to the translation-related molecules overlap each other, with centroid-to-centroid separation of 3.5060(11) Å between the prazole ring at (*x*, *y*, *z*) and triazine ring at (*x*, *y*, −1 + *z*) and triazine ring at (*x*, *y*, *z*) and pyrazole ring at (*x*, *y*, 1 + *z*). The *π*...*π* distances are 3.4112(8) and 3.3995(7) Å, respectively, and the angle between overlapping planes is 0.84(10)°.

### 2.5. Theoretical Calculations

Theoretical calculations provided the useful information about the molecular and electronic structures of the investigated compounds, and provided optimized molecular structures of **1**–**8** for molecular docking procedure. The electronic parameters (Table 4) were theoretically calculated at the DFT/B3LYP/311++G(d,p) level for compounds **1**–**8** in the conformation of the molecules corresponding to the minimum energy. The molecules **1**–**8** in conformation obtained after energy minimization and geometry optimization with vectors of dipole moment are shown in Figure 4.

Theoretical calculations showed that N-1-isomeric forms for compounds **2** and **4** have lower total energy than respective N-2-isomeric forms for **3** and **5**. Moreover, the N-2-substituted pyrazolo[4,3-*e*][1,2,4]-triazine derivatives are more polar than those substituted at N-1 position. The dipole moments of molecules **1**–**8** vary from 3.100 D for **1** to 8.420 D for **3.** The vectors of dipole moments have similar spatial orientation for all molecules being roughly parallel to the pyrazolo[4,3-*e*][1,2,4]-triazine system.

In order to characterize the reactivity and stability of investigated compounds, the energy of HOMO and LUMO orbitals as frontier molecular orbitals (FMOs) for molecules **1**–**8** were calculated. The concept of HOMO−LUMO energy gap is useful tool in predicting kinetic stability and chemical reactivity of the molecules [38,39].

The energies of HOMO, LUMO and energy gaps for all investigated compounds are presented in Table 4 and their graphical representation is shown in Appendix A. The energies of HOMO and LUMO orbitals for all investigated compounds are very close to each other, changing in the range of 7.75 and 5.94 kcal/mol for E_HOMO_ and E_LUMO_, respectively. Similarly, the energy gaps are within a very narrow range of 1.437 kcal/mol. This may indicate a similar reactivity of the tested compounds, and their similar behaviour under physiological conditions.

The net charge on the atoms in the 3-methyl–5-methylsulfanyl-1*H*-pyrazolo[4,3-*e*][1,2,4]triazine part of the molecules **1**–**8** calculated using NBO method is presented in Table 5. The atomic charges are very similar in all analysed molecules, however it can be seen that introduction of the substituent in N-1 or N-2 position of the pyrazolo[4,3-*e*][1,2,4]triazine rings system decreases the absolute value of negative atomic charge on N7 and N8 atoms, respectively, in comparison with the values of the net charge at these atoms observed in **1**. As expected, relatively large negative charges were observed at nitrogen and carbon atom of methylene groups, while large positive charges are present at sulphur and other carbon atoms.

### 2.6. Molecular Docking

The lack of significant anti-cancer activity of the investigated compounds and their structural similarity to the antiviral drug aciclovir prompted us to try to check in silico the antiviral activity of acyclonucleosides **1**–**8** using molecular docking procedure. The compounds **1**–**8** were tested in virtual screening to the thymidine kinase TK. This enzyme is responsible for the antiviral activity of aciclovir, converting it into a virus-infected cell to an active phosphate derivative that inhibits DNA synthesis and blocking the viral replication process [40]. In crystalline state, TK occurs in the complex with acyclovir, wherein its hydroxyethoxymethyl group has two possible locations in the binding site of TK. The crystal structure of TK with location of acyclovir molecule in active sites is showed in Appendix A [38].

The molecular docking results obtained for **1**–**8** and aciclovir molecule (**Ac**) are presented in Table 6. Graphical presentation of interactions between **1**–**8**, **Ac** and amino acids in the active site of TK enzyme is shown in Appendix A.

It can be noted that all analysed ligands bind to the active site of TK with the values of scoring function ChemPLP included in the narrow range from 43.24 for **1** (without acyclic side chain) and 53.74 for **7** to 64.74 for **8** and 77.94 for **Ac,** showing that all acyclonucleosides have similar affinity to TK enzyme. The best affinity to the binding site of TK exhibits ligand **8** which binds with ALA168A, TYR132A and ARG163A via C12…C, N8…O and O…N hydrogen bonds, respectively, involving atoms from both bicyclic part and acyclic chain of molecule (Figure 5a). In similar way as **8**, the molecule of **Ac** is linked to the active pocket of enzyme, but it should be noted that there are two possible position of the acyclic chain of **Ac** molecule in the crystal structure of TK (Figure 5b).

The molecular docking study showed that new obtained acyclonucleosides **2**–**8** can be considered as the compounds with potential antiviral activity by joining to the thymidine kinase and subsequently blocking the viral replication process.

## 3. Conclusions

In summary, new pyrazolo[4,3-*e*][1,2,4]triazine acyclonucleosides were designed and synthesized as a representative of new structural class of novel therapeutic agents. Using alkylation reaction we have produced a new series of prazolotriazine acyclonucleosides with different acyclic chains mimicking the sugar portion of naturally occurring nucleosides. The molecular structure of the model compound **3** was confirmed using X-ray analysis. Theoretical calculations performed at DFT/6-311++G(d,p) level showed that all investigated acyclonucleosides are characterized by similar reactivity and stability and they should therefore behave similarly under physiological conditions. The investigated compounds tested in vitro against human cancer cell lines (MCF-7, K-562) and CDK2/E as well as Abl protein kinases inhibitors showed no significant anticancer activity. However the structural similarity of acyclonucleosides **2**–**8** to antiviral drug aciclovir encouraged us to check in silico their antiviral activity using molecular docking procedure. The molecular docking of **1**–**8** to thymidine kinase TK revealed that all acyclonucleosides have similar affinity as aciclovir to active site of TK.

## 4. Experimental

### 4.1. Chemistry

All the reagents were analytical reagent or chemical pure. Melting points were determined on a Mel-Temp apparatus and are uncorrected. ^1^H- and ^13^C-NMR spectra were recorded on a Varian 400 MHz spectrometer (Siedlce University of Natural Sciences and Humanities, Siedlce, Poland,) (400 MHz for ^1^H and 100 MHz for ^13^C). The chemical shift values are expressed in ppm (part per million) with TMS as internal reference. The relative integrals of peak areas agreed with those expected for the assigned structures. Molecular weight of final compounds were assessed by electrospray ionization mass spectrometry (ESI/MS, Institute of Biochemistry and Biophysics, Polish Academy of Sciences, Warsaw, Poland) on a Agilent Technologies 6538 UHD Accurate Mass Q-TOF LC/MS. Elemental compositions are within ±0.4% of the calculated values.

General procedure for the preparation of derivatives **2**–**5**; A mixture of potassium carbonate (607 mg, 4.4 mmol) and derivative 1 (400 mg, 2.2 mmol) in anhydrous DMF (30 mL) was stirred for 10–15 min at room temperature. Then, a solution of appropriate alkylation agent: 2-bromoethanol or 3-bromo-1,2-propanediol (4.4 mmol) was added and stirred 3 days at room temperature. The reaction solution was poured over crushed ice (40 g) and the aqueous mixture extracted with diethyl ether (5 × 40 mL). The combined extracts were dried over anhydrous CaCl_2_ and evaporated under vacuum to give the required crude products. The crude products were submitted to column chromatography on silica gel using dichlorometane as eluent to give N1 alkylated derivative 2 or 4 as the first product. Further elution with CH_2_Cl_2_/EtOH (50:1) gives pure N2-alkylated derivative 3 or 5.

*1-(2-hydroxyethyl)-3-methyl–5-methylsulfanyl-1H-pyrazolo[4,3-e][1,2,4]triazine* (**2**), Yellow solid; Yield 65%, m.p. 140–143 °C. ^1^H-NMR (DMSO) δ: 2.55 (s, 3H), 2.67 (s, 3H), 3.88 (q, 2H, ^1^*J*_H-H_ = 5.6 Hz), 4.60 (t, 2H, ^1^*J*_H-H_ = 5.6 Hz), 4.81 (t, 1H, ^1^*J*_H-H_ = 5.6 Hz, OH exchanged with D_2_O); ^1^H-NMR (CDCl_3_) δ: 2.63 (s, 3H), 2.72 (s, 3H), 4.18 (t, 2H, ^1^*J*_H-H_ = 8.0 Hz), 4.74 (t, 2H, ^1^*J*_H-H_ = 8.0 Hz). ^13^C-NMR (CDCl_3_) δ: 10.93, 14.25, 50.79, 61.29, 135.04, 141.03, 147.32, 167.43. IR (KBr) cm^−1^: 873, 1070, 1395, 1433, 2924, 3403. HRMS (ESI, *m*/*z*) calc. for C_8_H_12_N_5_OS [M + H] 226.07571. Found 226.07542. Anal. Calcd. for C_8_H_11_N_5_OS: C, 42.65; H, 4.92; N, 31.09. Found: C, 42.52; H, 5.03; N, 31.00.

*2-(2-hydroxyethyl)-3-methyl-5-methylsulfanyl-2H-pyrazolo[4,3-e][1,2,4]triazine* (**3**), Yellow solid; Yield 10%; m.p. 176–178 °C; ^1^H-NMR (DMSO) δ: 2.61 (s, 3H), 2.64 (s, 3H), 3.90 (q, 2H, ^1^*J*_H-H_ = 5.6 Hz), 4.56 (t, 2H, ^1^*J*_H-H_ = 5.6 Hz), 5.00 (t, 1H, ^1^*J*_H-H_ = 5.6 Hz, OH exchanged with D_2_O); ^13^C-NMR (DMSO) δ: 8.79, 13.41, 54.49, 60.26, 128.52, 133.16, 154.20, 164.97. HRMS (ESI, *m*/*z*) calc. for C_8_H_12_N_5_OS [M + H] 226.07571. Found 226.07540. Anal. Calcd. for C_8_H_11_N_5_OS: C, 42.65; H, 4.92; N, 31.09. Found: C, 42.48; H, 5.10; N, 29.89.

*1-(2,3-dihydroxypropyl)-3-methyl–5-methylsulfanyl-1H-pyrazolo[4,3-e][1,2,4]triazine* (**4**), Yellow solid; Yield 27%; m.p. 150 °C; ^1^H-NMR (DMSO) δ: 2.48 (s, 3H), 2.59 (s, 3H), 3.45 (d, 2H, ^1^*J*_H-H_ = 5.2 Hz), 4.04 (m, 1H), 4.50 (m, 2 H). ^13^C-NMR (DMSO) δ: 10.64, 13.59, 51.56, 63.60, 70.11, 134.15, 139.29, 147.51, 165.64. HRMS (ESI, *m*/*z*) calc. for C_9_H_14_N_5_O_2_S [M + H] 256.08627. Found 256.08586. Anal. Calcd. for C_9_H_13_N_5_O_2_S: C, 42.34; H, 5.13; N, 27.43. Found: C, 42.22; H, 5.25; N, 27.32.

*2-(2,3-dihydroxypropyl)-3-methyl-5-methylsulfanyl-2H-pyrazolo[4,3-e][1,2,4]triazine* (**5**), Yellow solid; Yield 4%; m.p. 175 °C; ^1^H-NMR (DMSO) δ: 2.63 (s, 3H), 2.67 (s, 3H), 3.47 (t, 2H, ^1^*J*_H-H_ = 3.6 Hz), 4.04 (m, 1H), 4.48 (m, 1H), 4.60 (m, 1H), 4.96 (bs, 1H, OH), 5.19 (bs, 1H, OH); ^13^C-NMR (DMSO) δ: 8.16, 13.42, 54.88, 63.76, 71.70, 128.55, 133.37, 154.79, 165.63. HRMS (ESI, *m*/*z*) calc. for C_9_H_14_N_5_O_2_S [M + H] 256.08627. Found 256.08600. Anal. Calcd. for C_9_H_13_N_5_O_2_S: C, 42.34; H, 5.13; N, 27.43. Found: C, 42.18; H, 5.30; N, 27.22.

Reaction of **1** with dibromoethane; To a solution of **1** (1,5 g, 8,28 mmola) in DMF (60 mL), potassium carbonate (2.39 g, 0.017 mmola) was added and the resulting mixture was stirred for 10–15 min. at room temperature. Then, dibromoethane (3 mL, 0.044 mmol) was added and the reaction mixture was stirred for 3 days at room temperature. After that time, reaction mixture was poured into ice-water mixture and extracted with diethyl ether (5 × 40 mL). The combined organic extracts were dried over anhydrous CaCl_2_. After complete evaporation of the solvent, the residue was subjected to column chromatography eluting with CH_2_Cl_2_ to give **6a** as the first product. Further elution with CH_2_Cl_2_/hexane (5:1) gives pure **6** and **6b**.

*1-(2-bromoethyl)-3-methyl-5-methylsulfanyl-1H-pyrazolo[4,3-e][1,2,4]triazine* (**6**), Yellow solid; Yield 46%; m.p. 94 °C; ^1^H-NMR (CDCl_3_) δ: 2.62 (s, 3H), 2.72 (s, 3H), 3.87 (t, 2H, ^1^*J*_H-H_ = 6.4 Hz), 4.97 (t, 2H, ^1^*J*_H-H_ = 6.4 Hz); ^13^C-NMR (CDCl_3_) δ: 10.96, 14.25, 28.70, 49. 30, 135.11, 141.48, 147.31, 167.65; IR (KBr) cm^−1^: 553, 1428, 1547, 2923, 2968, 3404. HRMS (ESI, *m*/*z*) calc. for C_8_H_11_^81^BrN_5_S [M + H] 289.98926. Found 289.98887. Anal. Calcd. for C_8_H_10_BrN_5_S: C, 33.34; H, 3.50; N, 24.30. Found: C, 33.21; H, 3.49; N, 24.11.

*3-methyl-5-methylsulfanyl-1-vinyl-1H-pyrazolo[4,3-e][1,2,4]triazine* (**6a**), Yellow solid; Yield 7%; m.p. 88 °C; ^1^H-NMR (CDCl_3_) δ: 2.64 (s, 3H), 2.73 (s, 3H), 5.08 (dd, 1H, ^1^*J*_H-H_ = 8.4; 0.8 Hz), 6.09 (dd, 1H, ^1^*J*_H-H_ = 15.2, 0.8 Hz), 7.56 (dd, 1H, ^1^*J*_H-H_ = 15.6, 9.2Hz); ^13^C-NMR (CDCl_3_) δ: 10.94, 14.27, 101.04, 128.84, 136.41, 142.61, 146.21, 168.37; IR (KBr) cm^−1^: 627, 882, 1457, 1552, 1648, 2927; HRMS (ESI, *m*/*z*) calc. for C_8_H_10_N_5_S [M + H] 208.06514. Found 208.06505. Anal. Calcd. for C_8_H_9_N_5_S: C, 46.36; H, 4.38; N, 33.79. Found: C, 46.23; H, 4.47; N, 33.63.

*3-methyl-5-methylsulfanyl-2-vinyl-2H-pyrazolo[4,3-e][1,2,4]triazine* (**6b**), Yellow solid; Yield 5%; m.p. 197 °C; ^1^H-NMR (CDCl_3_) δ: 2. 66 (s, 3H), 2.72 (s, 3H), 5.50 (dd, 1H, ^1^*J*_H-H_ = 8.4, 0.8 Hz), 6.50 (dd, 1H, ^1^*J*_H-H_ = 15.2, 0.8 Hz), 7.37 (dd, 1H, ^1^*J*_H-H_ = 15.2, 8.8 Hz); ^13^C-NMR (CDCl_3_) δ: 8.83, 14.04, 111.69, 128.80, 129.64, 131.18, 154.48, 166.81. IR (KBr) cm^−1^: 852, 1429, 1511, 1631, 2923; HRMS (ESI, *m*/*z*) calc. for C_8_H_10_N_5_S [M + H] 208.06514. Found 208.06491. Anal. Calcd. for C_8_H_9_N_5_S: C, 46.36; H, 4.38; N, 33.79. Found: C, 46.45; H, 4.51; N, 33.60.

Synthesis of 1-(ethyl 2-carboethoxybutanoate-4-yl)-3-methyl-5-methylsulfanyl-1H-pyrazolo[4,3-e][1,2,4]triazine (**7**); Intermediate compound **6** (288 mg, 1 mmola) was dissolved in THF (40 mL). To the solution, diethyl malonate (0.8 mL, 0.851 g, 5.32 mmol) and NaH (414 mg, 50% dispersion in mineral oil) were added, followed by stirring for 48 h at room temperature. After completion of the reaction, the solvent was evaporated *in vacuo* and the crude product was submitted to column chromatography on silica gel using a mixture of CH_2_Cl_2_/hexane (50:1) as eluent to give derivative **7**; Yellow oil; Yield 60%; ^1^H-NMR (CDCl_3_) δ: 1.23 (t, 6H, ^1^*J*_H-H_ = 7.2 Hz), 2.59 (m, 5H), 2.70 (s, 3H), 2.71 (s, 3H), 3.34 (t, 1H, ^1^*J*_H-H_ = 14.8 Hz), 4.15 (m, 4H,) 4.69 (t, 2H, ^1^*J*_H-H_ = 6.4 Hz); ^13^C-NMR (CDCl_3_) δ: 10.89, 13.95, 14.21, 28.39, 45.73, 49.23, 61.68, 134.80, 140.81, 147.08, 167.28, 168.45; IR (KBr) cm^−1^: 657, 1494, 1546, 1732, 2983; HRMS (ESI, *m*/*z*) calc. for C_15_H_21_N_5_O_4_S [M] 367.13142. Found 367.13290. Anal. Calcd. for C_15_H_21_N_5_O_4_S: C, 49.03; H, 5.76; N, 19.06. Found: C, 48.87; H, 5.80; N, 18.87.

Reduction of **7**; Compound **7** (580 mg, 1,5 mmol) was dissolved in CH_2_Cl_2_ (8 mL), and MeOH (0,6 mL) and NaBH_4_ (330 mg, 9 mmol) were added. The reaction mixture was stirred at room temperature for 24 h, then diluted with water (8 mL) and extracted with CH_2_Cl_2_ (5 × 10 mL). The combined organic extracts were dried over anhydrous CaCl_2_. After complete evaporation of the solvent, the residue was subjected to column chromatography (CH_2_Cl_2_/ethyl alcohol, 100:1) to give **8** as the mine product and **8a**.

*1-[(4-hydroxy-3-hydroxymethyl)butyl]-3-methyl-5-methylsufanyl-1H-pyrazolo[4,3-e][1,2,4]triazine* (**8**), Yellow oil; Yield 36%; ^1^H-NMR (CDCl_3_) δ: 1.71 (m, 1H), 2.06 (q, 2H, ^1^*J*_H-H_ = 7.2 Hz), 2.54 (bs, 2H, 2 x OH), 2.60 (s, 3H), 2.69 (s, 3H), 3.78 (m, 4H), 4.69 (t, 2H, ^1^*J*_H-H_ = 7.2 Hz). ^13^C-NMR (CDCl_3_): 10.89, 14.19, 27.87, 39.54, 46.32, 64.76, 134.92, 140.58, 146.80, 167.14. HRMS (ESI, *m*/*z*) calc. for C_11_H_18_N_5_O_2_S [M + H] 284.11757. Found 284.11724. Anal. Calcd. for C_11_H_17_N_5_O_2_S: C, 46.63; H, 6.05; N, 24.72. Found: C, 46.70; H, 6.23; N, 24.65.

*[2- Hydroxymethyl-4-(3-methyl-5-methylsufanyl-1H-pyrazolo[4,3-e][1,2,4]triazin-1-yl]-butanoic acid ethyl ester* (**8a**), Yellow oil; Yield 7%; ^1^H–NMR (CDCl_3_) δ: 1.57–1.62 (m, 1H), 1.94 (bs, 1H), 2.08–2.12 (m, 2H), 2.62 (s, 3H), 2.72 (s, 3H), 3.67–3.70 (q, 2H, ^1^*J*_H-H_ = 13.0 Hz), 3.84–3.85 (t, 2H, ^1^*J*_H-H_ = 13.0 Hz), 4.64–4.68 (t, 2H, ^1^*J*_H-H_ = 14.4 Hz), 4.69–4.72 (m, 3H). ^13^C-NMR (CDCl_3_): 10. 93, 14.26, 28.09, 44.59, 48.04, 52.06, 61.07, 62.79, 134.86, 140.70, 147.03, 167.30. HRMS (ESI, *m*/*z*) calc. for C_13_H_20_N_5_O_3_S [M + H] 326.12814. Found 326.12785. Anal. Calcd. for C_13_H_20_N_5_O_3_S: C, 47.84; H, 6.18; N, 21.46. Found: C, 47.80; H, 6.32; N, 21.33.

### 4.2. Biological Studies

#### 4.2.1. Cytotoxicity Assays

The cytotoxicity of the studied compounds in K562 and MCF7 cell lines was determined as described earlier [29,30]. Briefly, the cells were assayed with compounds using three-fold dilutions in triplicate. Treatment lasted for 72 h, followed by addition of Calcein AM solution and measurement of the fluorescence of live cells at 485 nm/538 nm (excitation/emission).

#### 4.2.2. Kinase Inhibition Assay

Kinase assays were performed as described earlier [29,30] CDK2/cyclin E was assayed with 1 mg/mL histone H1 in the presence of 15 µM ATP, 0.05 µCi [γ-^33^P]ATP and of the test compound in a final volume of 10 µL in a reaction buffer (60 mM HEPES–NaOH, pH 7.5, 3 mM MgCl_2_, 3 mM MnCl_2_, 3 µM Na-orthovanadate, 1.2 mM DTT, 2.5 µg/50 µL PEG_20.000_). Abl was assayed with 500 µM of a synthetic peptide (GGEAIYAAPFKK) 10 µM of [γ -^33^P]ATP and the appropriate quantity of the test compound in a final volume of 10 µL. The reactions were stopped and spotted onto P-81 phosphocellulose (Whatman) and kinase inhibition was quantified using a FLA-7000 digital image analyzer (Fujifilm, Olomouc, Czech Republic).

### 4.3. Chromatographic Studies

#### 4.3.1. UV-Vis Spectroscopy

UV-Vis spectra of the compounds were recorded in water (20 mM sodium acetate buffer, pH 4; 20 mM ammonium acetate buffer, pH 7.4) - methanol (0.02 mg mL^−1^) (1:1, *v*/*v*)) solution on a UV-160A Shimadzu Spectrophotometer (Lublin, Poland). Quartz cuvettes (1 cm) were used for measurements.

#### 4.3.2. RP-18 Chromatography

A Eurosil Bioselect C18 (5 μm, 300 × 4.6 mm) column was used as the stationary phase. The mobile phase consisted of different volume mixtures of: (*i*) methanol and 20 mM ammonium acetate as the aqueous phase to give pH = 7.4; (*ii*) methanol and 20 mM sodium acetate buffer as the aqueous phase to give pH = 4. The methanol concentration ranged from 0.55 to 0.95 (*v*/*v*), depending on the structure of compound, at 0.05 intervals. The flow rate was 0.5 mL/min at room temperature. Measurements were made at 320 nm. The retention time of an unretained solute (t_0_) was determined by the injection of a small amount of acetone dissolved in water

#### 4.3.3. IAM Chromatography

A Rexchrom IAM.PC.DD2 (12 μm, 100 × 4.6 mm, 300 Å) (Regis Technologies) column was used as the stationary phase. The mobile phase consisted of different volume mixtures of acetonitrile and 20 mM ammonium acetate buffer as the aqueous phase to give pH = 7.4. The acetonitrile concentration ranged from 0.1 to 0.5 (*v*/*v*), depending on the structure of compound, at 0.05 intervals. The flow rate was 1 mL min^−1^ at room temperature. Measurements were made at 320 nm. The retention time of an unretained solute (t_0_) was determined by the injection of a small amount of citric acid dissolved in water.

HPLC measurements were carried out using a liquid chromatograph Knauer (Knauer, Berlin, Germany) with a dual pump and a UV–visible detector. The Clog P, log P and molar refractivity MR were calculated using the ChemDraw Ultra 10.0 according to the fragmentation method introduced by Crippen [36,41]. The polar surface area (tPSA) was estimated by the atom-based method using the MarvinViw 5.9.4 software [42,43].

### 4.4. X-ray Structure Determinations of **3**

X-ray data of (**3**) were collected on the Bruker SMART APEX II CCD diffractometer; crystal sizes 0.59 × 0.04 × 0.03 mm, CuK*α* (*λ* = 1.54178 Å) radiation, *ω* scans, *T* = 293 K, absorption correction: multi-scan SADABS [44], *T*_min_/*T*_max_ = 0.8131/1.0000. The structure was solved by direct methods using SHELXS97 [45] and refined by full-matrix least-squares with SHELXL97 [45]. The O-bound H atom was located by difference Fourier synthesis and refined freely. The remaining H atoms were positioned geometrically and treated as riding on their parent C atoms with C-H distances of 0.97 Å (CH_2_) and 0.96 Å (CH_3_). All H atoms were refined with isotropic displacement parameters taken as 1.5 times those of the respective parent atoms. Compound (**3**) crystallizes in non-centrosymmetric space group *Fdd*2 and the assumed absolute structure of its crystal was confirmed by the Flack parameter of 0.027(17) for 615 Friedel pairs [46]. All calculations were performed using WINGX version 1.64.05 package [47]. CCDC-1948479 contains the supplementary crystallographic data for this paper. These data can be obtained free of charge at www.ccdc.cam.ac.uk/conts/retrieving.html [or from the Cambridge Crystallographic Data Centre (CCDC), 12 Union Road, Cambridge CB2 1EZ, UK; fax: +44(0)-1223-336-033; E-mail: deposit@ccdc.cam.ac.uk].

Crystal data of (**3**): C_8_H_11_N_5_OS, *M* = 225.28, orthorhombic, space group *Fdd*2, *a* = 30.0070(9), *b* = 31.8941(1), *c* = 4.36030(10) Å, *V* = 4173.0(2) Å^3^, *Z* = 16, *d*_calc_ = 1.435 Mg m^−3^, *F*(000) = 1888, *μ*(Cu K*α*) = 2.634 mm^−1^, *T* = 293K, 11,154 measured reflections (*θ* range 4.05–69.30^o^), 1704 unique reflections, final *R* = 0.025, *wR* = 0.067, *S* = 1.047 for 1640 reflections with *I* > 2*σ*(*I*).

### 4.5. Theoretical Calculations

The energy and electronic parameters (frontier orbitals, dipole moments and NBO net charge distribution on the atoms) for all structures **1**–**8** were calculated with GAUSSIAN 03 [48] at the DFT/B3LYP level with 6-311++G(d,p) basis set. The initial geometries were built from the crystallographic data of **3** and the structures were fully optimized with energy minimisation. Calculations were carried out at the Academic Computer Centre in Siedlce. The visualization of theoretical calculation results was performed using GaussView [49]

### 4.6. Molecular Docking

The crystal structure of thymidine kinase TK in complex with aciclovir was downloaded from Protein Data Bank (PDB ID: 2KI5) [40]. Docking procedures were carried out for **1**–**8** using the GOLD Suite v. 5.7.2 software [50]. The enzyme preparation including addition of hydrogens, removal of water molecules, extraction of original ligand from the protein binding site, were done with GOLD default settings. The binding pocked of acyclovir molecule in crystal structure of TK was used as an active site with selection of atoms within 6 Å in molecular docking of investigated ligands. The acyclovir as reference ligand was removed from X-ray structure of its protein–ligand complex (2KI5) and docked back into its binding site with the RMSD values of 2.794 Å. In docking stimulations running with default parameters, each ligand was kept flexible but the amino acid residues of the enzyme were held rigid. ChemPLP scoring function was used to rank of docked ligands. Protein–ligand interactions were analysed using Hermes v. 1.10.2 [50].

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
