# Peer review of "Synthesis, Structural Characterization, and Biological Activity of New Pyrazolo[4,3-e][1,2,4]triazine Acyclonucleosides"

_molecules, 2020, doi:10.3390/molecules25010221_

Round 1

Reviewer 1 Report

Although there is no significant biological activity of the molecules. The manuscript is based on sound hypothesis and well-presented results. I recommend for publication in Molecules.

Author Response

Thank you very much for your review of our manuscript. 

Reviewer 2 Report

Attached you can find the pdf with some notes to clarify.

Furthermore, the biological activity of the synthesized compounds MUST be realized according to the title of the manuscript. The authors assayed only anticancer activity and this one is absent......so which kind of biological evaluation they mean?????

The HR_MS was made only for compound 7, WHY? The authors synthesized many compounds, and purity is important for all the compound.

This ms is only focused on the characterization of the synthesized compounds, so if the goal is the discovery of any new therapeutic application or biological activity, the correct biological assay must be carried out.

In the text, there are several mistakes.

Author Response

Obtained compounds were tested against cancer cell lines (K562 and MCF7) and as inhibitors of protein kinases CDK2/cyclin E and Abl. Moreover, molecular docking of obtained compounds 18 to thymidine kinase (TK) were conducted and revealed that all acyclonucleosides have similar affinity as aciclovir to active site of TK.  Therefore, in the title is  biological activity. HRMS were added to analytical data for compounds 2-6 and 8-8a.

Page 5, line 122

The sentence has been corrected and expanded:

The data show that some compounds possess a low distribution coefficient expressed by log P as well as by log kw determined on phosphatidylcholine as the stationary phase. It can be assumed that low lipophilicity of compounds (Clog P < -0.1) will not be conducive to the biological potency. This may limit absorption and distribution processes.

Reviewer 3 Report

This manuscript reports on the synthesis of series of new pyrazolo[4,3-e][1,2,4]triazine acyclonucleosides. Their anticancer activity against human cancer cell lines and Abl protein kinases inhibitors were investigated. . The structure of model compound was characterized by single crystal XRD. I believe it can be accepted with minor revisions:

There are too many editing or writing errors. (1) The axis on the UVv-vis spectra should be named (2) The IUPAC names of compounds, for example for compound 2 (1.-2-hydroxyethyl)-3-methyl–5-methylsulfanyl-1H-pyrazolo[4,3-e][1,2,4]triazine, p. 11 - 257) shouldn’t contain dots (see experiment part for all names) (3) mmola (p. 12 - 280) (4) to give 8 as the mine product and 8a. or ester etylowy kwasu, butanowego (mother language). The whole manuscript should be carefully checked and revised. The UV-vis spectra of all compounds should be presented in SI. The influenece of pH on the compound 1 requires explanation, for example the potassium pyrazolate salt can be obtained in KH2PO4 phosphate buffer (which buffer was used for pH 7.4?). What compound was investigated in this case at pH=7,4 for RP-18 and IAM chromatography. The coordinates for optimized geometries for all compounds should be presented. The ΔE for the compound 4 is given with mistake on fig 1S. “In order to characterize the reactivity and stability of investigated compounds the frontier 170 orbitals HOMO and LUMO, as the reactivity descriptors for molecules 18 were calculated” is a very controversial expression or literature reference should be given. What is the principal role of DFT calculations for this work? Why the H2O in DMSO-d6 signal possesses different positions in the NMR spectra p. 8 and 20 in SI? The names for all NMR spectra in SI are suggested to be given.

Author Response

The manuscript was revised again as regards English languages

The authors took into account the Reviewer's comments and performed chromatographic measurements at pH 7.4 using the ammonium acetate buffer (Tables 2 and 3). Perhaps the phosphate buffer was not the most optimal for testing the compounds under consideration.

According to the obtained results, their description was slightly modified. The UV-Vis spectra of all compounds were also made in water (sodium acetate buffer at pH 4; ammonium acetate buffer at pH 7.4) - methanol solutions (1:1, v/v). The C18 measurements at pH =4 were performed in the sodium acetate buffer. There was no effect of pH values  (in the range of 4-7.4) on the UV spectrum of the compounds.  For compound 1, no changes were registered either (Figure 2). When the phosphate buffer was used, some changes were observed for compound 1  (this may be the effect of buffer type) (the first version of the manuscript).

The analysis of the microspecies distribution of the compounds depending on the pH using numerical methods (Marvin) also indicates the lack of pH influence on the structure of the compounds. In the pH range of 4-7.4 the compounds exist in the solution in the molecular form. The UV-Vis spectrum of compounds 1 and 2 at pH 4 and 7.4 in acetate buffer was included in the revised  version of the manuscript (Figure 2). The other spectra were included in the supplementary materials  - according to the Reviewer’s remarks.

Figure 2 - The description of the coordinate axes was taken into account.

The coordinates for optimized geometries for all compounds should be presented.

The coordinates for optimized geometries for molecules of compounds 1 – 8 are presented in Supplementary Materials.

The ΔE for the compound 4 is given with mistake on fig 1S.

The error in ΔE for 4 in Fig. 1S is corrected. The ΔE value should be 83.740 kcal/mol.

“In order to characterize the reactivity and stability of investigated compounds the frontier 170 orbitals HOMO and LUMO, as the reactivity descriptors for molecules 1–8 were calculated” is a very controversial expression or literature reference should be given.

Page 11, line 7:

The phrase

“In order to characterize the reactivity and stability of investigated compounds the frontier 170 orbitals HOMO and LUMO, as the reactivity descriptors for molecules 18 were calculated”

is replaced by

“In order to characterize the reactivity and stability of investigated compounds the energy of HOMO and LUMO orbitals as frontier molecular orbitals (FMOs) for molecules 18 were calculated. The concept of HOMO−LUMO energy gap is useful tool  in predicting kinetic stability and chemical reactivity of the molecules [39, 40].”,

where

Chowdhury, S.; Bhuiya, S.; Haque, L.; Das. S. In-depth investigation of the binding of flavonoid taxifolin with bovine hemoglobin at physiological pH: Spectroscopic and molecular docking studies. Spectrochim. Acta A 2020, 225, 117513. Fleming, I. Frontier Orbitals and Organic Chemical Reactions, Wiley, London, 1976.

What is the principal role of DFT calculations for this work?

Page 9, part 2.5. Theoretical calculation:

The phrase is added

“Theoretical calculations gave the useful information about the molecular and electronic structures of the investigated compounds and provided optimized molecular structures of 18 for molecular docking procedure.”

Round 2

Reviewer 2 Report

The paper is amended 

This manuscript is a resubmission of an earlier submission. The following is a list of the peer review reports and author responses from that submission.

Round 1

Reviewer 1 Report

Mojzych et al describe a series of new pyrazolo[4,3-e][1,2,4]triazine acyclonucleosides were synthesized and evaluated for their anticancer activity  as well as for kinases inhibition. While reading the manuscript I commented directly in de attached pdf file.

Major concern:

Apart from the synthesis there is actually only incremental data on the biological activity of the molecules. What worries me most about the manuscript is the sentence: "Further development of this series of compounds and their biological activities will be reported in the future in the separate paper." But why a separate paper if you envision more biological data? Why the compounds were modeled in TK but not tested in TK? Why the panel of cancer cells was limited to two and not expanded to contain at least a representative cancer cell line for each tissue or representative for the most prevalent cancers?

Minor see attached pdf. Issues like compound numbering, some typos need to be resolved to improve the manuscript.

Reviewer 2 Report

The paper discusses the synthesis, the biological evaluation and the molecular docking of a new series of acyclonucleosides bearing a pyrazolo[4,3-e][1,2,4]triazine moiety. The work is conceptually well organized and the way the authors describe methods and used techniques is direct and simple. However, there are some critical points that compromise the publication of the article, at least in the current form:

The compounds do not show any protein kinase inhibition activity in micromolar range. This is the main issue of the work which therefore lacks in the target-based design. I suggest some structural modification (with the input of the docking studies) and see if there is something detectable in terms of activity. The supplementary materials document does not show any copy of the 1HNMR and 13CNMR of the compounds; at least, NMR spectra of the final products should be included. The synthetic pathways described, concern often low yield reactions, and I suggest you to improve the synthetic methods, optimizing the reaction steps. Experimental section should be revised, I suggest some modifications: coupling constant should be written like “1JH,H”, chemical formulas should be represented uniquely (if you use “CH2Cl2“ once, do not switch to “dichloromethane”), days should be changed in hours (3 days à 72 h). The overall grammatical structure of the text should be revised. There are lots of typing errors.